# Exploring the Effect of Sn Addition to Supported Au Nanoparticles on Reducible/Non-Reducible Metal Oxides Supports for Alkane Oxidation

Marta Stucchi [1], Alessandro Vomeri [1], Sándor Stichleutner [2], Károly Lázár [2], Emanuela Pitzalis [3], Claudio Evangelisti [3,*] and Laura Prati [1,*]

1 Dipartimento di Chimica, Università Degli Studi di Milano, Via Golgi 19, 20133 Milano, Italy; marta.stucchi@unimi.it (M.S.); alessandro.vomeri@unimi.it (A.V.)
2 Centre for Energy Research, Nuclear Analysis and Radiography Department, Institute for Energy Security and Environmental Safety, Konkoly-Thege M. Street 29-33, H-1121 Budapest, Hungary; stichleutner.sandor@ek-cer.hu (S.S.); lazar.karoly@ek-cer.hu (K.L.)
3 CNR—ICCOM— Institute of Chemistry of OrganoMetallic Compounds, Via G. Moruzzi 1, 56124 Pisa, Italy; emanuela.pitzalis@pi.iccom.cnr.it
* Correspondence: claudio.evangelisti@cnr.it (C.E.); laura.prati@unimi.it (L.P.)

**Abstract:** Acetone-stabilized Au- and Sn-solvated metal atoms (SMAs) were used as to obtain Au- and AuSn-supported catalysts by simple impregnation on a reducible ($TiO_2$) and a non-reducible ($Al_2O_3$) metal-oxide. Their catalytic behaviour was investigated for cyclohexane oxidation to cyclohexanol and cyclohexanone (KA oil), and their morphological and physical properties were studied by TEM, STEM-EDS and $^{119}Sn$-Mössbauer spectroscopy. The catalytic results firstly demonstrated that the bare supports played a role on the reaction mechanism, slowing down the formation of the oxidation products and directing the radical formation. Hereinafter, the comparison between the monometallic Au-supported catalysts and the corresponding bimetallic Au-Sn catalysts allowed for the understanding of the potential role of Sn. $^{119}Sn$-Mössbauer characterization analyses showed the presence of $SnO_2$, which was recognized to favour the electrons' exchange to form radicals, interacting with oxygen. Such interaction, in particular, could be favoured by the co-presence of Au. Moreover, the same metal composition on the catalyst surface resulted in a different catalytic behaviour depending on the support.

**Keywords:** KA oil; Au-Sn-supported catalyst; solvated metal atom dispersion; SMAD; cyclohexane oxidation; Mössbauer spectroscopy; $SnO_2$; Au-Sn/$Al_2O_3$; Au-Sn/$TiO_2$

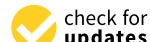



## 1. Introduction

Supporting gold nanoparticles have shown to be extremely active for many industrially important reactions, including oxidations [1]. Catalytic oxidations can be classified into complete oxidation, used for the catalytic degradation of toxic compounds, and selective oxidation, used for organic compounds in fine chemistry. Gold catalysts proved to be efficient for both types of oxidations, including the oxidation of CO [2], the hydrogen production by water–gas shift (WGS), the oxidative decomposition of volatile organic compounds (VOCs) [3] and selective oxidation of alcohols, hydrocarbons and sugars [4–7]. However, from an industrial point of view, the increasing interest in environmental issues put the attention on the need for more efficient processes, new methods for the synthesis of nanoparticles and the rational use of non-critical metals.

Concerning the metal active species, Au showed normally high selectivity [6] but lower activity than other precious metals. This limitation can be overcome by the use of a second metal which can improve catalytic activity (synergistic effect) and selectivity. The study of the effect of the addition of non-noble and abundant metals would be also recommended

considering the cost and restraint of noble metals (such as gold). Recent studies are thus more focused on the exploration of cheap, non-toxic and environmentally friendly catalysts. Among some bio-relevant and earth-abundant metals, tin can be rather interesting [8]. Tin is also poorly investigated as an active component of heterogeneous catalysts, while it has been more reported in the field of electrocatalysis. For example, it has been reported to be a good transition metal for the development of affordable electrocatalysts for the reduction of $CO_2$ to CO [9] as well as for the reduction of $CO_2$ to hydrocarbons or to formic acid [10]. Moreover, Sn has been reported to be used for the electro-oxidation of ethanol [11,12], in particular coupled with Pt [13], where carbon-supported Pt/Sn catalysts were prepared both by decorating carbon-supported Pt with Sn and by the co-deposition of Pt and Sn on carbon. The coupling of Sn with Pt was also recently reported for the base-free oxidation of HMF [14], where Pt and Pt/Sn nanoparticles (NPs) were synthesized using Pt carbonyl cluster precursors deposited on $TiO_2$. The subsequent addition of Sn to Pt in that case was demonstrated to lead to the formation of unique active sites with significantly improved stability in the base-free oxidation of HMF.

In this work, Au and Au-Sn-supported catalysts were synthesized using Au and Sn metal powders as precursors by the solvated metal atom dispersion (SMAD) method [15,16]. Acetone-stabilized Au- and Sn-solvated metal atoms (SMAs) were used to obtain Au- and AuSn-supported catalysts by simple impregnation on a reducible ($TiO_2$) and a non-reducible ($Al_2O_3$) metal-oxide. The morphological and physical properties of the samples were characterized by TEM, STEM-EDS and $^{119}$Sn-Mössbauer spectroscopy, and their catalytic behaviour was studied for cyclohexane oxidation. It is known that the C–H bond(s) of hydrocarbons, in particular alkanes, can be transformed by oxidation reactions into C-OH or C=O groups to obtain highly added value products which often have applications in fine chemistry. However, the activation of the C–H bond still remains a challenge [17,18]. In particular, the oxidation of cyclohexane to cyclohexanol (CyOH) and cyclohexanone (Cy=O) (K–A oil) (Figure 1) has great industrial importance considering that the products are important compounds to be used in the production of ε-caprolactam and adipic acid, which is used in the nylon-6 and nylon-6,6 polymers manufacture [19].

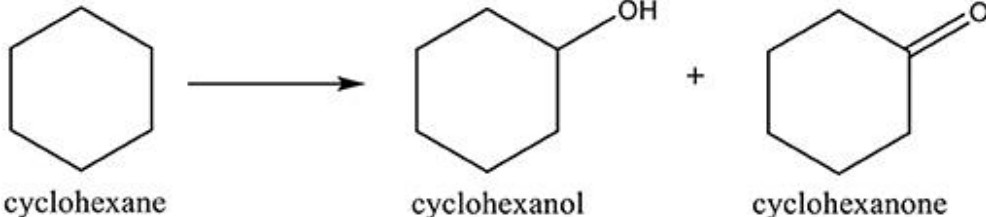

**Figure 1.** Reaction scheme of the cyclohexane oxidation to KA oil.

The selective oxidation of cyclohexane to cyclohexanone and cyclohexanol has already been reported using Au-supported catalysts [20–22] and using a peroxide initiator such as TBHP (tert-Butyl hydroperoxide). Recently, we demonstrated the possibility to use, as a radical initiator, the auto-oxidation of benzaldehyde [23]. The yield in K–A oil was maximized using a bimetallic 2%Au-Cu/$Al_2O_3$ catalyst, but some fundamentals of the reaction mechanism still need to be clarified. For example, it has been controversially debated if Au acts as a catalyst or as an initiator of the radical reaction [24]. In the literature, we found both explanations: for example, Lai et al. [25] reported that 5 nm of Au nanoparticles is very effective in activating molecular oxygen, which is the rate-determining step in aerobic cyclohexane oxidation; on the contrary, Weckhuysen et al. [26] reported that cyclohexane oxidation in the presence of gold-based catalysts is not a catalytic process but proceeds simply through a radical-chain mechanism. Nevertheless, the use of a Au/MgO catalyst suggested an intermediate scenario, where Au could accelerate the reaction even without an initiator (thus behaving like a real catalyst), but the acceleration took place when the amount of some species was increased, i.e., $C_6H_{11}$–OOH or $C_5H_{11}$–OO•, which promoted the catalytic processes by a radical chain mechanism [27].

Considering all of this, a deep understanding of the reaction mechanism could be very helpful in a further optimization of the reaction. It should be noted that the bimetallic structures used in this work allowed us to clarify some important aspects of this mechanism.

## 2. Materials and Methods

### 2.1. Materials

$Al_2O_3$ (Degussa (Zürich, Switzerland), Aluminium oxide, SA: 100 m$^2$ g$^{-1}$) and $TiO_2$ (P25 by Degussa) were chosen as supports. The acetone was from Merck; it was distilled and stored under argon. Gold (beads, 1–6 mm, 99.999%) and Tin (shot, 3 mm, 99.999%) were purchased from Sigma-Aldrich (St. Louis, MO, USA). The acetone-solvated metal atoms solutions were handled in an argon atmosphere with the use of the standard Schlenk technique. Benzaldehyde (ReagentPlus®, ≥99%, Sigma-Aldrich) and Cyclohexane (ACS reagent, ≥99%, Sigma-Aldrich) were purchased and used without further purification.

### 2.2. Catalysts Synthesis

The synthesis of Au mono- and AuSn catalysts was carried out in a previously described reactor, following the solvated metal atom dispersion method (SMAD) [15]. Monometallic Au catalysts were prepared by condensation of Au (100 mg, beads) vapour with acetone vapours (100 mL). The resulting Au-Acetone-solvated metal atoms (SMAs) (96 mL) containing 0.70 mg/mL of Au (measured by the ICP-OES analysis) were divided into two portions (48 mL) that were adsorbed on 1.0 g of the $Al_2O_3$ and $TiO_2$ support, respectively. The mixtures were stirred for 12 h at room temperature. The supernatant solutions were then removed, and the red-brown solid was washed with *n*-pentane (3 × 50 mL) and dried under reduced pressure.

For the preparation of the bimetallic AuSn catalysts, Au and Sn vapours generated at $10^{-5}$ mbar by the resistive heating of two different alumina-coated tungsten crucibles (filled with 100 mg of Au beads and 12 mg Sn shot) were co-condensed simultaneously with acetone (100 mL) onto the cold walls of the glass reactor and maintained at liquid nitrogen temperature (−196 °C) for 1 h. The reactor chamber was heated to the melting point of the solid matrix (cca. −95 °C), and the resulting acetone-stabilized Au-Sn-solvated metal atoms (SMAs) solution (96.0 mL) was kept in an argon atmosphere in a Schlenk tube at −80 °C. The resulting Au-Sn/Acetone-solvated metal atoms (SMAs) (96 mL) containing 0.65 mg/mL of Au and 0.11 mg/mL of Sn were divided into two portions (48 mL) that were added to an acetone suspension of 1.0 g of the $Al_2O_3$ and $TiO_2$ support, respectively. The mixtures were stirred for 12 h at room temperature. The supernatant solutions were then removed and the red-brown solid was washed with *n*-pentane (3 × 50 mL) and dried under reduced pressure. A similar procedure was used to prepare the Au-Sn catalysts containing a different Au/Sn molar ratio, using 96 mL of a Au-Sn/acetone SMA, containing 0.78 mg/mL of Au and 0.62 mg/mL of Sn and 1.7 g of the $Al_2O_3$ and $TiO_2$ support, respectively.

All the prepared Au- and AuSn-supported systems were mineralized using a microwave digester, and the residual solution was examined by ICP-OES. The actual metal (Au + Sn) wt. % total content with the molar ratio between the two metals was calculated by ICP analyses (Table 1). In the last column, the mean particle diameter of the Au-containing particles (when applicable) was evaluated by TEM analyses, as has been reported.

**Table 1.** Nominal and actual Au/Sn molar ratio, metal wt. % and mean particle diameter from TEM analyses.

| Sample | Au/Sn Molar Ratio | | Metal Wt. (%) | | Mean Particle Diameter(nm) (TEM) |
|---|---|---|---|---|---|
| | Nominal | actual | nominal | actual | |
| **Au$_3$Sn$_1$/TiO$_2$** | 3 | 2.91 | 3 | 3.29 | - |
| **Au$_3$Sn$_1$/Al$_2$O$_3$** | 3 | 2.74 | 3 | 3.30 | - |
| **Au$_1$Sn$_2$/TiO$_2$** | 0.5 | 0.54 | 3 | 3.26 | 6.28 ± 3.11 |
| **Au$_1$Sn$_2$/Al$_2$O$_3$** | 0.5 | 0.50 | 3 | 3.35 | 7.24 ± 3.56 |

The choice of having a specific ratio between the Au and Sn metals came from the previous findings using these ratios, whereby it is possible to obtain stable alloys [28].

### 2.3. Catalysts Characterization

An ICP-OES Perkin Elmer optical emission spectrometer, Optima 8000 ICP-OES, was used to evaluate the actual metal loading of each catalyst. The samples were dissolved using a CEM MARS One Microwave Digester, and the digestion was performed at 200 °C using a mixture of HCl (37%), $H_2SO_4$ (95%) and $HNO_3$ (65%) with a ratio of 1:3:1.

$^{119}$Sn Mössbauer spectra were recorded on a KFKI spectrometer, operating in constant acceleration mode, in transmission geometry at 78 K and 298 K using an in situ cell [29] with a 555 MBq $Ca^{119m}SnO_3$ source. The velocity scale was calibrated with $\alpha$-Fe measurements, while the isomer shifts are given relative to $SnO_2$. The accuracy of the positional parameters is $\pm0.03$ mm s$^{-1}$. The recorded spectra were evaluated by least-square fitting of the lines using the MossWinn code [30]. A Lorentzian line shape was used for the decomposition; no parameters were constrained.

An FEI Titan Themis 200 kV spherical aberration (Cs)—corrected TEM with 0.09 nm HRTEM and 0.16 nm STEM resolution equipped with 4 Thermofischer (Waltham, MA, USA) EDS detectors—was applied to investigate the morphology and composition of the AuSn bimetallic samples drop-dried from aqueous suspensions onto Cu microgrids. The mean diameter of Au-containing particles was calculated by measuring more than 260 particles. The composition of the samples was determined by STEM-EDS along with recording HAADF (high-angle annular dark-field) images on selected areas. The data were evaluated, and Au and Sn elemental maps were created by background-corrected and -fit Au-L or Sn-L intensities using the Velox software.

### 2.4. Oxidation of Cyclohexane

The oxidation of cyclohexane (10 mL) was carried out in a 100 mL stainless-steel autoclave at 120 °C and 4 bars of oxygen pressure using undecane as the external standard (0.2 M) and 20 mg of the catalyst. In accordance with our previous work [23], the reaction was performed in the presence of benzaldehyde (0.15 M). Stirring was set to 1100 rounds per minute. Oxygen was refilled after each withdraw of the sample at t = 0, 30 min, 1 h, 3 h, 4 h and 5 h (restoring the internal pressure at 4 bars of $O_2$). The reactor was cooled down in an ice bath each time to withdraw a sample. For the characterisation of the reaction products, samples were centrifuged to separate the catalyst from the solution, and the liquid solution was analysed using a GC (Thermo Scientific TRACE 1300) equipped with an Agilent HP-5 column.

### 3. Results and Discussion

#### 3.1. Catalysis

3.1.1. Cyclohexane Oxidation in the Presence of Bare Supports

The reaction of cyclohexane oxidation in the presence of benzaldehyde proceeds even without a catalyst (Figure 2a) [23]. In this case, it is expected to have a purely radical mechanism, where the ketone is formed in excess with respect to the alcohol (K/A ratio = 1.7) [23,31].

Rather surprisingly, the addition of the bare supports, i.e., $Al_2O_3$ (Figure 2b) or $TiO_2$ (Figure 2c), seems to slow down the formation of the oxidation products with respect to the reaction without catalysts, perhaps influencing the radical formation. This latter hypothesis is in fact supported by some studies, which described the possible role of $Al_2O_3$ as a radical scavenger [32,33]. For example, Fukuzumi et al. [33] reported the effects of $Al^{3+}$ on the scavenging of a 2,2-diphenyl-1-picrylhydrazyl radical (DPPH·), which is frequently used as a reactivity model of peroxyl radicals' behaviour. In that case, the study proved that $Al^{3+}$, as a strong Lewis acid, can act as a radical-scavenger by stabilising the one-electron-reduced species of the radical [33]. Still more recently, Jabbari et al. [34] described the adsorption of the free radical TEMPO ($C_9H_{18}NO$) on $Al_2O_3$ nanoparticles in different

solvents, evaluating the radical scavenging activity. It was proven that $Al_2O_3$ nanoparticles have efficient radical scavenging activity (RSA) in the range of 50–72% as well as a large adsorption energy associated with the adsorption over the Lewis acidic sites of $Al_2O_3$ [35].

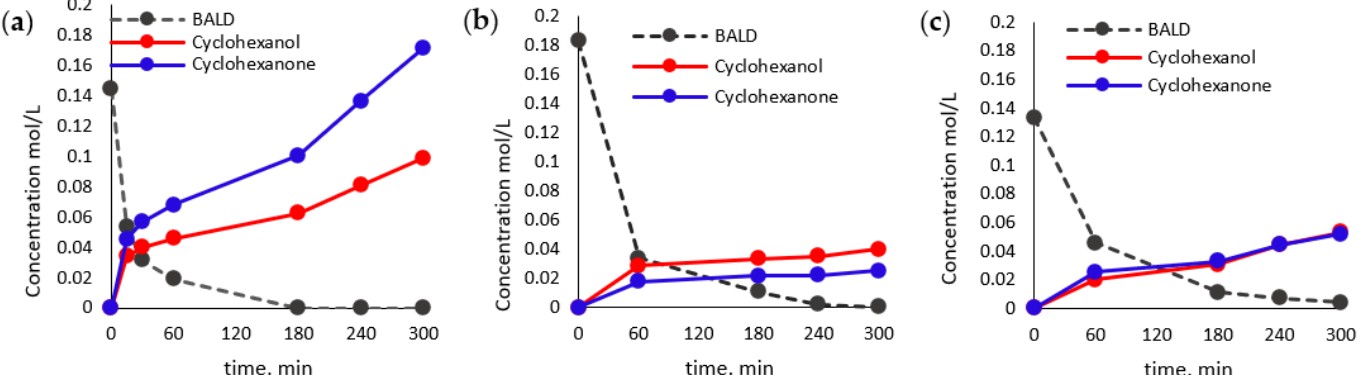

**Figure 2.** Cyclohexane (10 mL) oxidation in the presence of benzaldehyde (0.15 M) at 120 °C and 4 bars of $O_2$. Reaction performed without a catalyst (**a**), in the presence of 20 mg of $Al_2O_3$ (**b**) and 20 mg of $TiO_2$ (**c**).

The explanation of the role of the bare $TiO_2$ on the radical mechanism of the reaction seems to be more difficult. Indeed, $TiO_2$ is usually considered to be able to produce reactive radical species (e.g., radical $^\bullet OH$, $O_2^{\bullet-}$), which are generated at its surface. However, this happens only under UV irradiation [36,37]. On the other hand, both the Lewis and Brönsted acid sites have been recognized on the surface of $TiO_2$-based materials by the adsorption of organic dyes or 4-carboxy TEMPO [38]. From these data, considering the proved adsorption of radical species on the Lewis acid sites of $Al_2O_3$, we suppose a similar behaviour on the Lewis acid sites of $TiO_2$, which can be the reason on the basis of the reduced reaction rate of the K–A oil formation in the presence of $TiO_2$ compared to the non-catalysed reaction.

It should be noted that in the reactions performed using the bare supports, the K/A ratio is also lower than that of the non-catalysed reaction (Table 2). This is another signal of the inhibition of the radical pathway.

**Table 2.** Catalytic performance of each sample as K–A formation and related K/A ratio.

| Entry | | Cy–OH (mmol) | Cy=O (mmol) | K–A Oil (mmol) | K/A |
|---|---|---|---|---|---|
| 1 | *No cat.* | *0.99* | *1.72* | *2.70* | *1.74* |
| 2 | $TiO_2$ | 0.53 | 0.52 | 1.05 | 0.98 |
| 3 | $Au/TiO_2$ | 1.70 | 1.48 | 3.18 | 0.87 |
| 4 | $Sn/TiO_2$ | 0.45 | 0.34 | 0.79 | 0.76 |
| 5 | $Au_3Sn_1/TiO_2$ | 1.49 | 1.51 | 2.99 | 1.01 |
| 6 | $Au_1Sn_2/TiO_2$ | 1.46 | 1.68 | 3.14 | 1.15 |
| 7 | $Al_2O_3$ | 0.40 | 0.25 | 0.65 | 0.63 |
| 8 | $Au/Al_2O_3$ | 1.92 | 1.74 | 3.66 | 0.90 |
| 9 | $Sn/Al_2O_3$ | 0.70 | 0.38 | 1.08 | 0.54 |
| 10 | $Au_3Sn_1/Al_2O_3$ | 1.08 | 1.07 | 2.15 | 0.99 |
| 11 | $Au_1Sn_2/Al_2O_3$ | 1.06 | 1.15 | 2.21 | 1.09 |
| 12 | $Au/Al_2O_3$ + $Sn/Al_2O_3$ | 1.28 | 0.79 | 2.07 | 0.62 |

### 3.1.2. Cyclohexane Oxidation in the Presence of Monometallic Au-Supported and Sn-Supported Catalysts

Considering the Au-based catalysts, the addition of Au nanoparticles on the supports' surface increased the catalytic activity (Figure 3). The K–A productivity (mmol) increased from 2.70 mmol of the non-catalysed reaction to 3.18 mmol in the presence of 3%Au/$TiO_2$

and to 3.66 mmol with 3%Au/Al$_2$O$_3$ (Table 2). Moreover, the effect of the catalyst was marked, considering the selectivity to the alcohol or the ketone formation: in both of the catalysed reactions, the amount of the alcohol is higher than the ketone, which is exactly the opposite of what happens when only the radical mechanism is active.

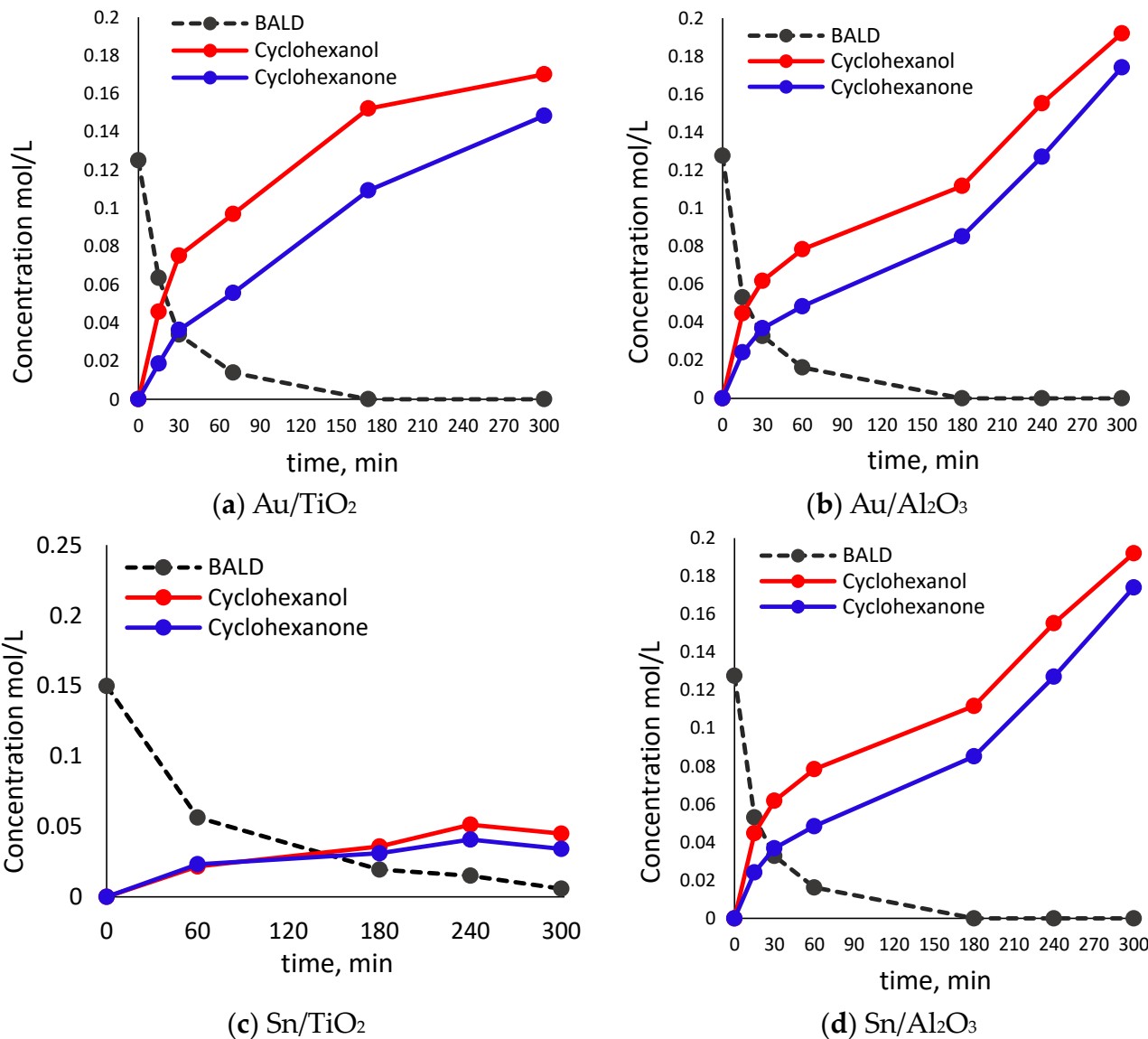

**Figure 3.** Cyclohexane (10 mL) oxidation in the presence of benzaldehyde (0.15 M) at 120 °C and 4 bars of O$_2$. Reaction performed in the presence of 20 mg of Au/TiO$_2$ (**a**); 20 mg of Au/Al$_2$O$_3$ (**b**); 20 mg of Sn/TiO$_2$ (**c**); 20 mg Sn/Al$_2$O$_3$ (**d**).

Indeed, when the reaction is ruled by a pure radical mechanism, the K/A ratio is 1.74 (Table 2). The larger formation of ketone compared to the alcohol is in agreement with the previous literature reports when a catalyst is present [31,39]. On the contrary, the presence of Au active sites in the reaction reversed the ratio (0.87–0.90), increasing the formation of the alcohol.

However, this behaviour seems to be independent of the support since the K/A ratio values obtained with TiO$_2$ or Al$_2$O$_3$ are very similar (Table 2).

In addition, it is possible to notice a slight difference in the reaction profiles, where the Au/Al$_2$O$_3$ catalyst did not show a deactivation, different from Au/TiO$_2$ where the reaction profile seems to go toward a plateau. This aspect will be better discussed later, together with the results obtained with the Au-Sn bimetallic catalysts.

Differently, in the case of monometallic Sn-based catalysts, the presence of Sn did not change the catalytic behaviour of the bare supports a lot, which showed, in fact, a similar productivity and K/A oil ratio. Overall, the KA oil productivity obtained with the monometallic Sn-supported catalysts was lower than that obtained with the corresponding Au-based catalysts.

### 3.1.3. Cyclohexane Oxidation in the Presence of AuSn-Supported Catalysts

When Sn was present and in intimate contact with Au (bimetallic catalysts), the final formation of the K–A oil increased with respect to the monometallic Sn catalysts, while it decreased with respect to the monometallic Au catalysts (see Table 2. However, the effect of Sn is mostly marked, regarding the formation of the alcohol or the ketone (Figure 4). It is in fact immediately clear from the reaction profiles that the K/A ratio is always close to 1 (Figure 4 and Table 2, last column), different from all the other cases.

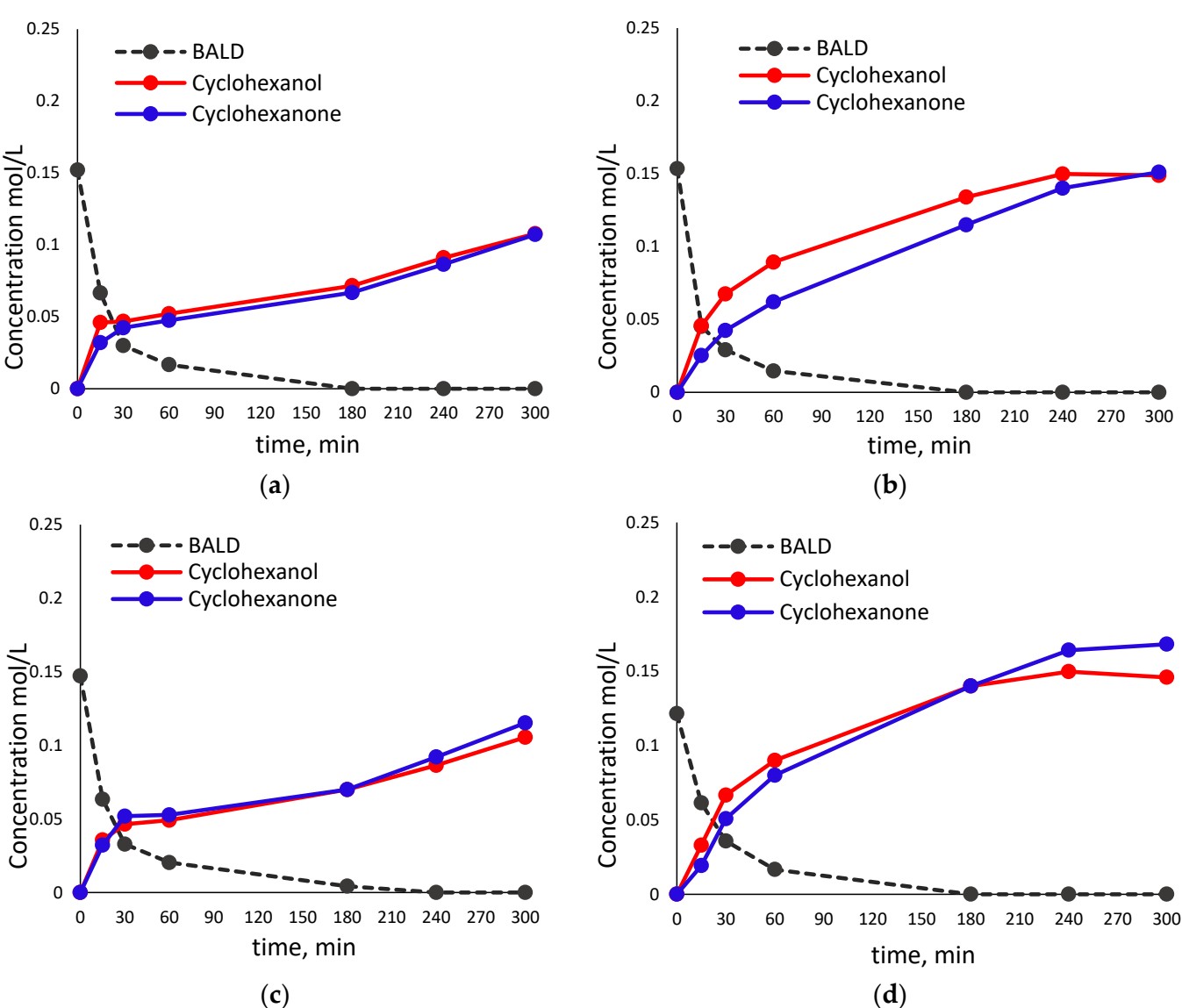

**Figure 4.** (**a–f**) Reaction profiles of cyclohexane oxidation in the presence of bimetallic AuSn catalysts supported on $Al_2O_3$ or $TiO_2$, respectively. Reaction conditions: 120 °C, 4 bars of $O_2$, benzaldehyde, 0.15 M, as radical initiator and 20 mg of catalyst. (**a**) $Au_3Sn_1/Al_2O_3$; (**b**) $Au_3Sn_1/TiO_2$; (**c**) $Au_1Sn_2/Al_2O_3$; (**d**) $Au_1Sn_2/TiO_2$.

For this reason, it can be supposed that the co-presence, in intimate contact with Au and Sn, influences the radical mechanism and the relative formation of radical species in a different way. To confirm this hypothesis, a reaction was performed in the presence of a physical mixture of $Au/Al_2O_3$ and $Sn/Al_2O_3$ (Figure 5).

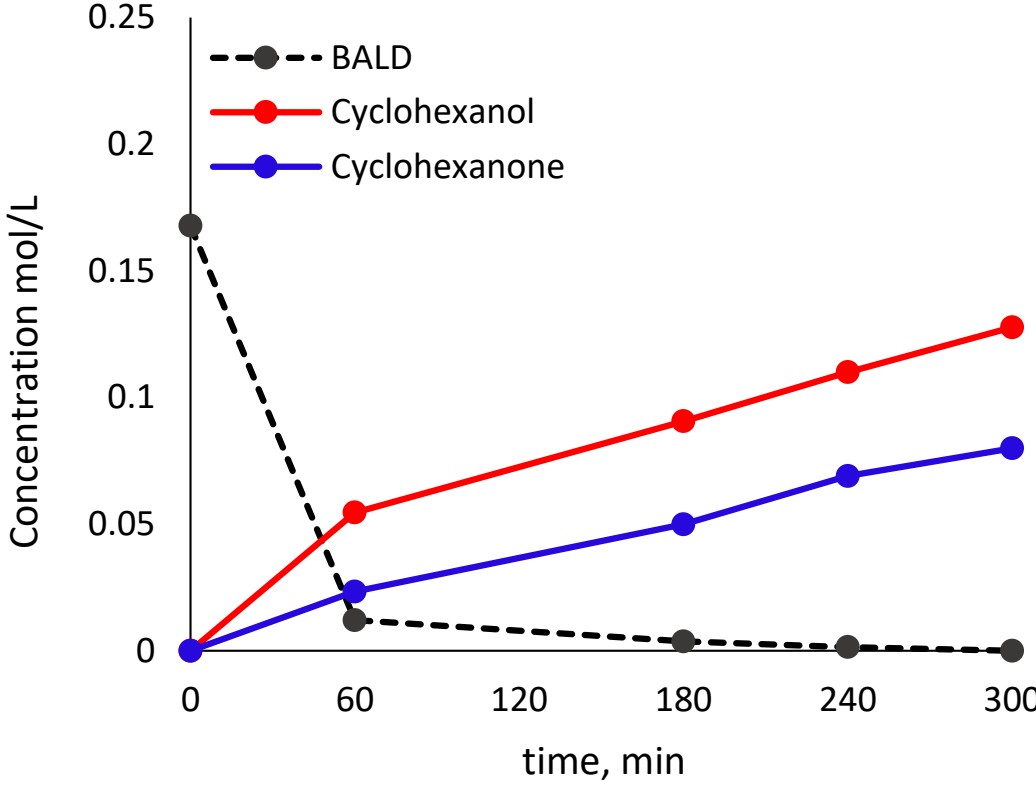

**Figure 5.** Cyclohexane (10 mL) oxidation in the presence of benzaldehyde (0.15 M) at 120 °C and 4 bars of $O_2$. Reaction performed in the presence of a physical mixture of $Au/Al_2O_3$ and $Sn/Al_2O_3$.

Looking at the total mol of the KA oil formed at the end of the reaction (Table 2, 5th column), first of all, it can be noted that those formed using a physical mixture of $Au/Al_2O_3$ + $Sn/Al_2O_3$ (entry 12) are lower than those obtained by the bimetallic $Au_1Sn_2/Al_2O_3$ (entry 11). Moreover, crucial for understanding the role of Sn and Au, the K/A ratio is severely different, changing from 1.09 in the case of the bimetallic catalyst to 0.62 using the physical mixture of the monometallic samples.

In addition, looking at the reaction profiles of the bimetallic samples (Figure 4a–f), some observations that highlight a possible support effect can be performed: indeed, it can be noted that in each case where $Al_2O_3$ is the support (Figure 5a,c,e), the reaction profiles exhibited no deactivation; on the contrary, the curves of those where $TiO_2$ is the support (Figure 5b,d,f) go towards a plateau, which can indicate that the catalyst is gradually decreasing its activity. We supposed a "combined effect", depending both on the support and the metal species, considering that this difference was not present in the bare supports (see Figure 2).

### 3.2. Characterization

#### $^{119}$Sn-Mössbauer Spectroscopy and TEM, STEM-EDS of AuSn Catalysts

A total of 298 and 78 K of the $^{119}$Sn Mössbauer spectra were collected from the more tin-rich $Au_1Sn_2/Al_2O_3$ and $Au_1Sn_2/TiO_2$ samples to investigate the chemical state of tin. The probability of the Mössbauer effect depends on the temperature of the measurement; it is more expressed for zerovalent tin (and bimetallic alloy) components than for Sn(IV). The characteristic isomer shift (IS) values for Sn(IV) are around 0.0 mm s$^{-1}$. The isomer shift of Au-Sn alloys depends inversely on the composition; it is 2.56 mm s$^{-1}$ for the metallic

β-Sn and, ca., 2.0 mm s$^{-1}$ for the gold-rich substitutional alloys [37]. Tin usually prefers to spread over the oxide supports, and indeed dedicated methods should be used to obtain true bimetallic AuSn particles [38].

The recorded spectra are presented in Figure 6. The data obtained from their deconvolutions are listed in Table 3 Spectra of the alumina-supported sample were collected with better measuring statistics, providing means to resolve two Sn(IV) components.

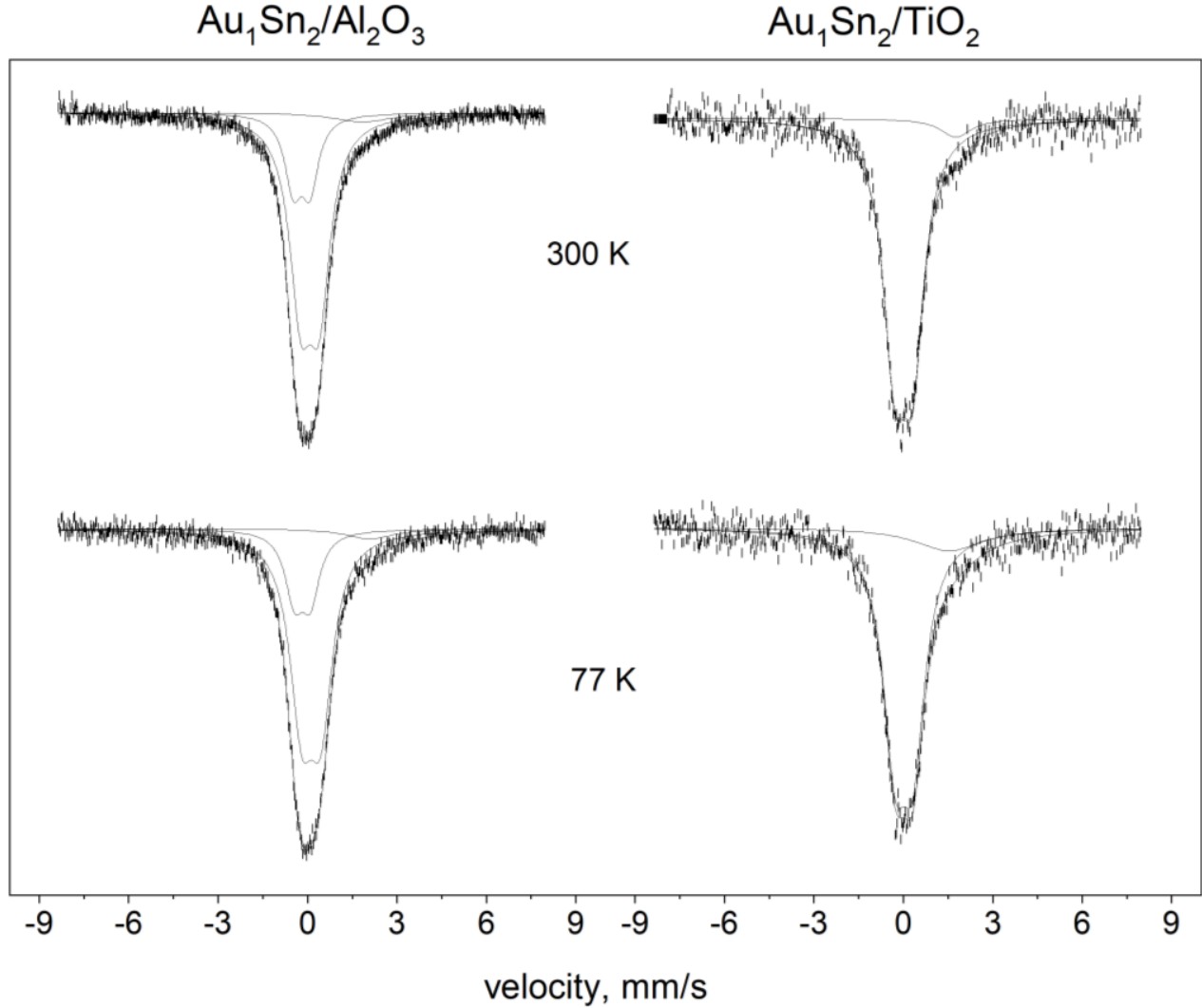

**Figure 6.** $^{119}$Sn Mössbauer spectra of the $Au_1Sn_2/Al_2O_3$ and $Au_1Sn_2/TiO_2$ samples.

**Table 3.** Mössbauer parameters of spectra presented in Figure 6.

| Temp. | | 300 K | | | | 77 K | | | |
|---|---|---|---|---|---|---|---|---|---|
| Support | Component | δ [a] | Δ [b] | FWHM [c] | Rel. Int. [d] | δ | Δ | FWHM | Rel. Int. |
| | | mm s$^{-1}$ | mm s$^{-1}$ | mm s$^{-1}$ | % | mm s$^{-1}$ | mm s$^{-1}$ | mm s$^{-1}$ | % |
| **Al₂O₃** | Sn(IV)-(1) | −0.20 | 0.56 | 0.69 | 22.4 | −0.17 | 0.54 | 0.75 | 21.9 |
| | Sn(IV)-(2) | 0.08 | 0.62 | 0.90 | 72.5 | 0.12 | 0.65 | 0.97 | 73.7 |
| | singlet | 1.84 | — | 2.35 | 5.1 | 2.08 | — | 1.91 | 4.4 |
| **TiO₂** | Sn(IV) | 0.00 | 0.60 | 0.93 | 94.3 | 0.02 | 0.61 | 0.98 | 86.2 |
| | singlet | 1.76 | — | 1.28 | 5.7 | 1.56 | — | 2.72 | 13.8 |

[a] δ, isomer shift, [b] Δ quadrupole splitting, [c] full line width at half maximum, [d] spectral contribution.

The spectra attest to the overwhelming presence of Sn(IV) components. This observation is in good correspondence with the affinity of tin to oxygen especially in well-dispersed form. It should be noted that the metallic tin oxidizes, during storage in air, if it is not well stabilized in an alloy.

A broad additional singlet contribution is also present in each sample with low intensity. The isomer shift of this component is definitely below 2.0 mm s$^{-1}$: the starting value characterizing the low tin-content Au-Sn bimetallic particles. The line width of this component is broad, probably indicating that different types of local tin species are buried beneath. Most probably, this component can be attributed to the tin in the Au-rich substitutional AuSn alloy.

The TEM images present (Figure 7) well-distributed metal particles of 7.2 ± 3. 6 and 6.3 ± 3.1 nm mean particle diameters with a relatively large size range on both the alumina and titania supports, respectively. The dispersion of the metals is slightly larger on TiO$_2$ than on Al$_2$O$_3$.

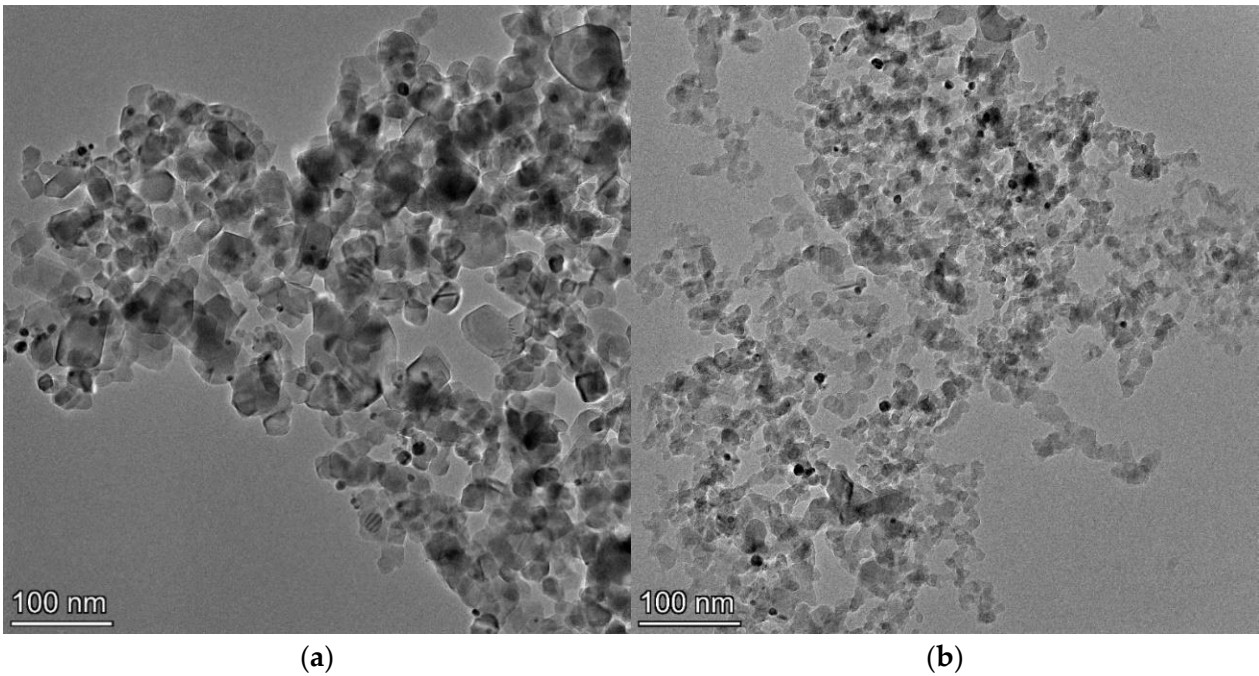

(**a**) (**b**)

**Figure 7.** Representative TEM images of (**a**) Au$_1$Sn$_2$/TiO$_2$ (d$_m$ = 6.28 ± 3.11 nm) and (**b**) Au$_1$Sn$_2$/ Al$_2$O$_3$ (d$_m$ = 7.24 ± 3.56).

The interaction and morphology of the active components were investigated by STEM-EDS and visualized on Au and Sn elemental maps (Figure 8—Au$_1$Sn$_2$/TiO$_2$—and Figure 9—Au$_1$Sn$_2$/Al$_2$O$_3$). Gold is concentrated in particles, while tin is more dispersed. The Au-containing particles are clearly visible as bright contrast features in the HAADF images and appear in red in the elemental maps. According to the green regions in the elemental maps, the Sn-containing phases appear more distributed on the support and must be SnO$_2$, as suggested by Mössbauer spectroscopy, giving a hardly recognizable and faint contrast in the HAADF images. The interaction of SnO$_2$ seems to be more intense with titania (Figure 8) than alumina (Figure 9), since on the latter, it is more granular, while it is more spread on the former support. The combined Au and Sn maps show that Au particles are typically surrounded by Sn-containing phases, and the joint appearance of Sn and Au signals at single particles likely indicates the presence of Au-Sn alloyed particles, in accordance with the Mössbauer spectroscopy results or the Au particles decorated by SnO$_2$.

Based on these structural investigations of both Au$_1$Sn$_2$ catalysts, only low amounts of metallic Sn in interaction with gold in the Au-rich alloy could be detected, while the largest part of tin, 95% on Al$_2$O$_3$ and 86% on TiO$_2$, is present as Sn(IV), likely as SnO$_2$. The

Au(Sn) particles are typically decorated, surrounded by $SnO_2$, probably diminishing the concentration of the accessible surface Au sites. Another significant part of Sn-oxide is located on the support separated from gold.

We have to remind ourselves that cyclohexane oxidation, in the presence of AuSn catalysts, produced higher amounts of ketone with respect to the monometallic Au catalysts (K/A > 1—Table 2). The K/A ratio in the presence of Sn approaches the value obtained when the pure radical pathway is active, thus letting us suppose that Sn favours this mechanism.

Indeed, as in the case of the two supports used ($TiO_2$ and $Al_2O_3$), $SnO_2$ has cations with empty orbitals and charge-deficient anions [40], which could favour the electrons' exchange to form radicals. Consider also the interaction of $SnO_2$ with oxygen. There is a very recent paper [41] which showed that upon adsorption of $O_2$ molecules on $SnO_2$ surfaces, superoxide ($O_2^-$) ions, singly ($O^-$) and doubly ($O^{2-}$) ionized atomic oxygen or also peroxide ($O_2^{2-}$) ions can be formed [41]. In particular, what is very interesting is the difference between the adsorption of $O_2$ on $SnO_2$ or $SnO_2$ containing one extra electron: in the first case, the $O_2$ molecules maintain their O–O bond, while for the $O_2$ molecules on $SnO_2$ containing one extra electron, the chemisorption is evident from the formation of two separate Sn–O bonds, with a formation of $O_2^-$ (proved by the energetic shifting of the antibonding $\pi *$ orbital). Our hypothesis is that the co-presence of Au (as the metal) and Sn (as the oxide) could favour the electron exchange, improving the radical pathways on the catalyst surface in the presence of oxygen. The Mössbauer analyses showed, in fact, the presence of $SnO_2$ both in the case of the Alumina and $TiO_2$ supports, and the same catalytic behaviour, in terms of the K/A ratio, was found.

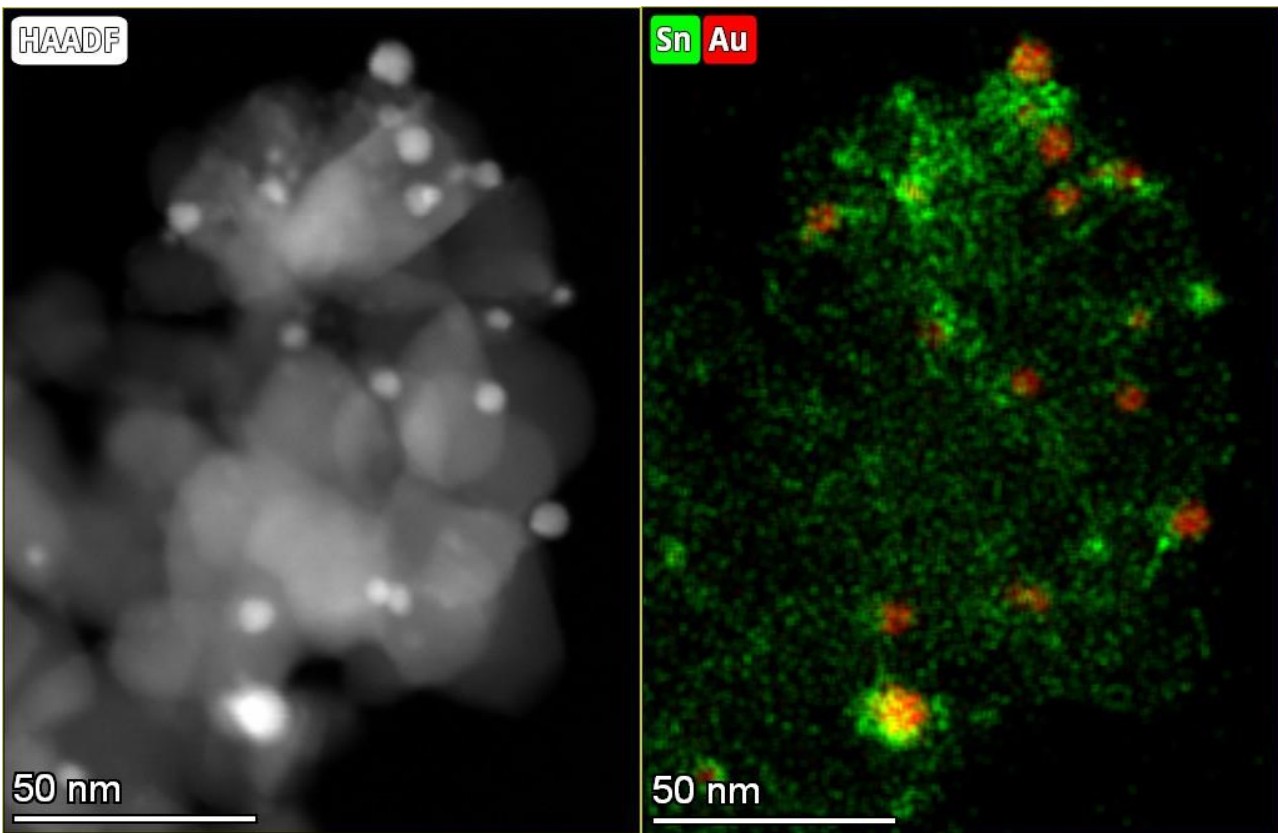

**Figure 8.** Au and Sn elemental maps with HAADF image of $Au_1Sn_2/TiO_2$ (on this selected area, Au/Sn = 32/68).

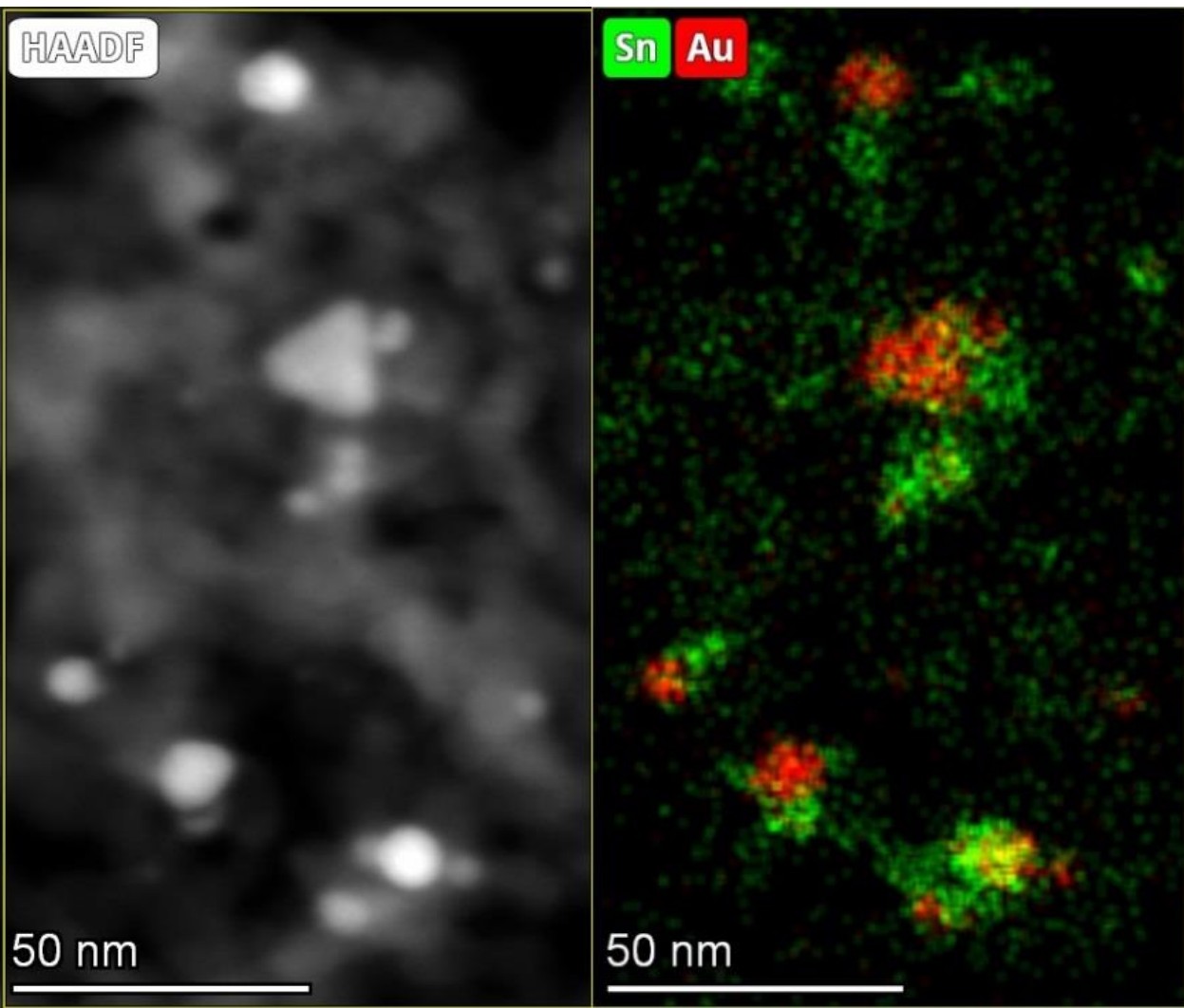

**Figure 9.** Au and Sn elemental maps with HAADF image of Au$_1$Sn$_2$/Al$_2$O$_3$ (on this selected area, Au/Sn = 45/55).

Instead, the catalytic behaviour seemed to be different considering the activity and the stability of the different supported catalysts. Indeed, TiO$_2$-supported catalysts showed a final higher productivity than the Al$_2$O$_3$-supported ones, but their reaction profiles have a tendency to plateau, which can be an indication of a slow deactivation. On the contrary, the reaction profiles of the Al$_2$O$_3$-supported catalyst showed a slower reaction rate but without any sign of deactivation.

Looking at the TEM analyses (Figure 7), a first explanation of this slightly different activity can be found in the different metal particle size, which was lower for the TiO$_2$ samples and higher for the Al$_2$O$_3$ samples, and the accessible Au surface might be different due to the dissimilar coverage by SnO$_2$. Moreover, the different supports led to a different dispersion of SnO$_2$, which could be a reason of the higher stability of the Al$_2$O$_3$-supported catalysts.

## 4. Discussion of the Mechanism

According to the information obtained from the catalytic tests and characterization analyses, some discussion about the reaction mechanism can be conducted. Scheme 1 reports an exemplification of the free radical pathways occurring in the oxidation of cyclohexane. During the initiation stage of the reaction, the activation of the C–H bond of cyclohexane (CyH) can occur by the abstraction of an H atom, resulting from the cleavage

by an unsaturated metal centre, a peroxide species or a superoxide species bound to metal centres or metal oxides (1a) [27]; otherwise, the chain initiation can start from the catalytic oxidation of CyH to cyclohexyl hydroperoxide (CyOOH) (1b) [42]. From the subsequent destruction of the radical species, cyclohexanol (CyOH) and cyclohexanone (Cy=O) can be formed (see steps 5–8).

(1a) $CyH + X \rightarrow Cy^\bullet + X\text{-}H$

or

(1b) $CyH + O2 \, (+cat) \rightarrow CyOOH$

Initiation

(2) $Cy^\bullet + O2 \rightarrow CyOO^\bullet$

(3) $CyOO^\bullet + CyH \rightarrow CyOOH + Cy^\bullet$

(4) $CyOOH \rightarrow CyO^\bullet + {}^\bullet OH$

(5) $CyO^\bullet + CyH \rightarrow CyOH + Cy^\bullet$

(6) $CyOO^\bullet + CyOOH \rightarrow CyOOH + Cy(^\bullet)OOH$

(7) $Cy(^\bullet)OOH \rightarrow Cy{=}O + OH^\bullet$

Propagation

(8) $2\,CyOO^\bullet \rightarrow CyOH + Cy{=}O + O_2$

Termination

**Scheme 1.** Exemplification of the free radical pathways occurring in the oxidation of cyclohexane.

In this context, the superoxide ($O_2^-$) ions, singly ($O^-$) and doubly ($O^{2-}$) ionized atomic oxygen or peroxide ($O_2^{2-}$) ions, forming from the interaction between the molecular $O_2$ and $SnO_2$, could play a critical role (Figure 10). However, the formation of superoxides occurs particularly in the presence of one extra electron, which is the reason why the co-presence of Au nanoparticles is crucial, as proven by the catalytic tests using the monometallic Sn-catalyst or the physical mixture of Au- and Sn-supported samples (see Figures 3 and 5), which both present a lower activity than the AuSn bimetallic catalyst.

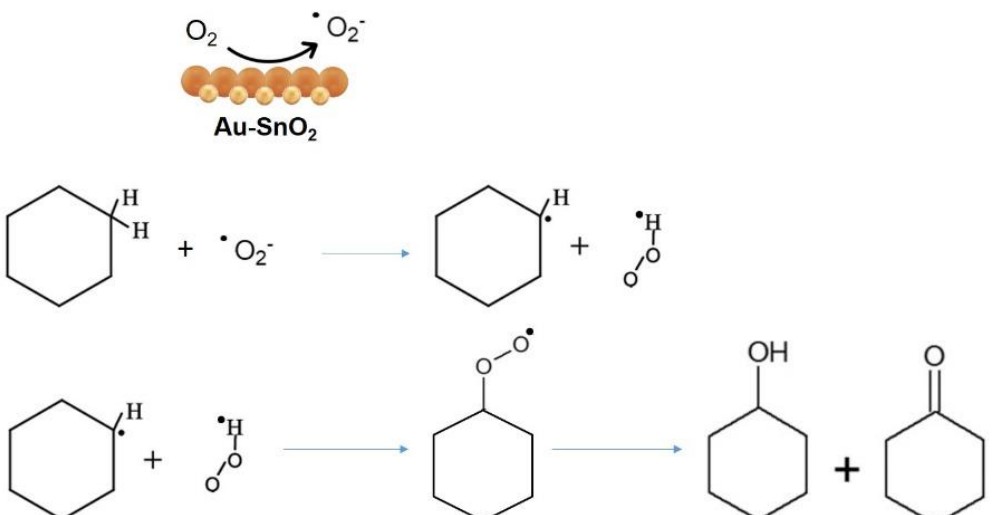

**Figure 10.** Generation of superoxide ($O_2^-$) ions from molecular oxygen and mechanism of cyclohexane oxidation to KA oil.

From here, the proposed mechanism, which occurs in the presence of AuSn-supported catalysts, would start, with the $O_2$ splitting into peroxide radicals, which subsequently activate cyclohexane, forming its radical species.

This hypothesis is supported by the catalytic results of the reaction performed using the $Au_1Sn_2/Al_2O_3$ catalyst, at 120 °C and 4 bars of $O_2$ in the absence of a radical initiator (Table 4). As indicated in Table 4, after 2 h of the reaction, both cyclohexanol and cyclohexanone were formed.

**Table 4.** KA oil formation after 2 h of cyclohexane (10 mL) oxidation in the presence of 20 mg of $Au_1Sn_2/Al_2O_3$ at 120 °C and 4 bars $O_2$ and in the absence of a radical initiator.

| Time (min) | Cyclohexanol (mmol) | Cyclohexanone (mmol) |
|:---:|:---:|:---:|
| 0 | 0 | 0 |
| 120 | $0.93 \times 10^{-3}$ | $1.5 \times 10^{-3}$ |

## 5. Conclusions

This work presented an effective synthesis methodology to prepare Au-Sn-supported catalysts, obtained using Au and Sn metal powders as precursors by the solvated metal atom dispersion (SMAD) method. The supports were selected to compare a reducible metal-oxide support ($TiO_2$) and a non-reducible one ($Al_2O_3$). The catalytic behaviour was studied for cyclohexane oxidation, considering the still open challenge in the activation of the C–H bond and the importance that a thorough understanding of the mechanism behind this reaction would have.

The catalytic results firstly showed that the bare supports, i.e., $Al_2O_3$ or $TiO_2$, also played a role in the reaction mechanism, slowing down the formation of the oxidation products. $Al_2O_3$ was assessed to act as the radical scavenger, particularly due to the presence of $Al^{3+}$ as the Lewis acid sites; similarly, in the case of $TiO_2$, the possible presence of both Lewis and Brönsted acid sites could be the reason for the reduced reaction rate of K–A oil formation.

The comparison between the monometallic Au-supported catalysts, Sn-supported catalysts and the corresponding bimetallic Au-Sn catalysts has allowed for the understanding of the potential role of Sn, which did not improve the catalytic activity in terms of the productivity of the K–A oil (mmol/h) (compared to the monometallic Au catalysts) but resulted in a different K/A ratio. The pure radical mechanism resulted in a K/A ratio of 1.74, while the presence of Au active sites reversed the ratio to 0.87–0.90; differently, the addition of Sn made the ratio closer to 1. Considering that this happened only when Au

and Sn were in intimate contact, we ascribed these results to the possible enhancement of the radical pathway related to the presence of $SnO_2$, revealed by the analyses of the $^{119}Sn$-Mössbauer spectra: indeed, $SnO_2$ has cations with empty orbitals and charges deficient anions, which could favour the electrons' exchange to form radicals; moreover, there could be a particular interaction of $SnO_2$ with oxygen, from which superoxide ($O_2^-$) ions, singly ($O^-$) and doubly ($O^{2-}$) ionized atomic oxygen or also peroxide ($O_2^{2-}$) ions can be formed, but such interaction occurs only in the co-presence of Au (as the metal) and Sn (as the oxide).

As a last observation, the same metal composition on the catalyst surface resulted in a different catalytic behaviour depending on the support, because the $Al_2O_3$-supported catalysts showed no deactivation, different from $TiO_2$. Such difference could come from the different metal particle size, which was lower for the $TiO_2$ samples and higher for the $Al_2O_3$ samples, together with the different availability of Au sites due to the dissimilar coverage by $SnO_2$.

These results disclosed a very interesting role of $SnO_2$ in contributing to the process of radicals' formation resulting from oxygen splitting, which is a key aspect for this type of oxidation reactions. The effect seems to be enhanced thanks to the presence of Au, where the co-presence of the metal species can favour the electron exchange and therefore the radicals' formation. Therefore, from here, the exploration of other metals, especially transition metals, coupled with Sn will be very promising for the development of even more sustainable and active catalysts.

**Author Contributions:** Conceptualization, M.S. and A.V.; methodology, M.S. and A.V.; validation, M.S., L.P. and C.E.; formal analysis, S.S., K.L. and E.P.; investigation, M.S. and A.V.; resources, L.P. and C.E.; data curation, M.S. and C.E.; writing—original draft preparation, M.S.; writing—review and editing, M.S., C.E. and L.P.; supervision, L.P.; project administration, L.P and C.E.; funding acquisition, L.P and C.E. All authors have read and agreed to the published version of the manuscript.

**Funding:** Bilateral project agreement CNR-MTA 2019-2022.

**Data Availability Statement:** Not applicable.

**Acknowledgments:** The authors are greatly thankful to Andrea Beck and György Sáfrán (Centre for Energy Research, Institute of Technical Physics and Materials Science) for their great help in the characterization analyses. The Authors gratefully acknowledge CNR and HAS(MTA) for the support in the frame of the bilateral project agreement CNR-MTA 2019-2022 andof the Hungarian Scientific Research Fund OTKA (Grant numbers K143216).

**Conflicts of Interest:** The authors declare no conflict of interest.

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
