# Peer review of "Exploring the Effect of Sn Addition to Supported Au Nanoparticles on Reducible/Non-Reducible Metal Oxides Supports for Alkane Oxidation"

_chemistry, doi:10.3390/chemistry5030107_

Round 1

Reviewer 1 Report

The overall logic of the thesis is clear, the research method is scientific and reasonable, and the innovation lies in the presented an effective synthesis methodology to prepare Au-Sn supported catalysts, obtained using Au and Sn metal powders as precursors by the solvated metal atoms dispersion (SMAD) method, And the conclusions you draw in the text are supported by ample data.

There are some problems, which should be solved before it is considered for publication.

First of all, Some discussion of the mechanism needs to be added to your article so that the reader can gain a deeper understanding of the article. I suggest that the section "Discussion of the mechanism" be added before "CONCLUTION" in the article.

Secondly, the clarity of some of the images in your article could be spruced up and improved, and the formatting of some tables should be adjusted appropriately. the paper's grammar and academic expression are rough. It is recommended to further enhance the clarity of the picture and polish the language, otherwise, it will be difficult to attract readers.

You should read more literature to expand on the “INTRODUCTION” of this article so that it is of interest to the reader. It is recommended to include "Green Energy & Environment, 2022. doi:10.1016/j.gee.2022. 01.005" in your article.

Given a satisfactory response from the authors, I would propose that the paper be accepted.

English needs to be properly embellished. 

Author Response

We thank the Reviewer for all the comments, questions and indication.

In the word file attached, a point-by-point response to the comments is provided.

Reviewer 2 Report

This paper is titled “Exploring the effect of Sn-doped Au nanoparticle catalysts on the reaction activity of cyclohexane oxidation to produce K-A oil.” The simultaneous loading of Au and Sn on an oxide support by SMAD method seems to be aimed at alloying Au and Sn. However, as shown in Fig. 9, only a small amount of Sn was solidly soluble in Au in the catalyst, with about 90% or more present as SnO4. Furthermore, the authors state that the independent SnO4 promotes a "pure radical pathway". These suggestions indicate that it does not make sense to alloy Sn with Au, and the need to use the SMAD method is unclear. Also, if the effect of independently present SnO4 is to be verified, there must be a catalyst impregnated with Au and Sn simultaneously or sequentially as a comparison catalyst that is completely free of Au-Sn alloys.

Looking at the activity data in Table 2, the highest K-A oil formation is observed for Au/Al2O3 and the highest K/A ratio is observed for No cat. and no significance can be found for Au-Sn composites. Even if the aim for this experiment is to reduce the K/A ratio, Al2O3 is the best, and there is no significance in Au-Sn compositing.

From the above points, it is unclear what the paper is trying to claim, and if the conclusion is that independently existing SnO4 promotes the "pure radical pathway," then the above comparative catalyst data are essential.

The authors should examine the above comments thoroughly, conduct experiments with additional comparative catalysts, and reorganize the paper to clarify the point of appeal. In doing so, the title should naturally be changed.

Author Response

We thank the reviewer for the comments, suggestions and questions.

In the word file attached, a point-by-point response to the comments is provided.

Reviewer 3 Report

The present paper is focused on the Sn addition effect to supported Au nanoparticles on Al2O3 or TiO2 catalyzing alkane oxidation. In spite of several novel findings, it cannot be accepted for the publication, unless it will be revised majorly. See the following comments.

1.     In order to determine whether the reaction is a non-catalytic radical reaction, a radical reaction on the catalyst surface, or a normal catalytic reaction that starts with adsorption, product distribution (activity and selectivity change) alone is not enough. is.

The reaction system should be a differential one to eliminate the initial concentration dependence of the reactants.

At least, you should change the reaction conditions, such as changing the reaction temperature, and observe the changes in the product distribution. Furthermore, if the reaction path changes, the activation energy will also change, so this should also be properly quantified.

It is only after careful analysis in this way that it is possible to discuss reaction pathways.

2.     The role of Sn is also not so simple. At least the effect of Sn addition cannot be made clear unless 4 to 6 types of catalysts with different amounts of Sn addition are tested. The lack of data for only Sn supported catalysts decisively undermines the discussion.

3.     In order to show the superiority of this catalyst preparation method, comparison with the usual Au and Sn loading method is essential. The usual supporting methods are the impregnation one and/or the ion exchange one.

Author Response

We thank the Reviewer for all the comments, suggestions and questions.

In the word file attached, a point-by-point response to the comments is provided.

Round 2

Reviewer 2 Report

Thank you for your careful revision of the previous manuscript, including the additional experiments and the addition of relevant discussions.

The reviewer confirmed that the revisions including the title change were acceptable, and that the appeal points and the conclusions of the paper was concise and clear.

This manuscript will be recommended to this journal.

Reviewer 3 Report

The revised manuscript seems well improved and then acceptable for the publication.